# Exploring the Moderating Role of Social and Ethical Practices in the Relationship between Environmental Disclosure and Financial Performance: Evidence from ESG Companies

**Salim Chouaibi [1], Matteo Rossi [2,*] , Dario Siggia [3] and Jamel Chouaibi [1]**

1  Department of Accounting, Faculty of Economics and Management of Sfax, University of Sfax, Sfax 3018, Tunisia; salim.chouaibi@fsegs.usf.tn (S.C.); chouaibi_jamel@yahoo.fr (J.C.)
2  Department of Law, Economics, Management and Quantitative Methods (DEMM), University of Sannio, 82100 Benevento, Italy
3  Department of Agricultural and Forest Sciences, University of Palermo, 90128 Palermo, Italy; dario.siggia@unipa.it
*  Correspondence: mrossi@unisannio.it

**Abstract:** Environmental disclosure is the latest novelty in the corporate reporting field. In fact, it is a tool that can better represent the capacity of companies in creating financial performance over time. Therefore, this paper analyzes whether environmental disclosure (ED) practiced by firms listed on the ESG index affects their financial performance (FP) using the moderating effect of social and ethical practices. The analysis includes a linear regression using panel data from Thomson Reuters and Bloomberg databases. Panel data were collected from a sample of 523 companies listed on the North American and West European stock exchanges. The obtained results show a positive and significant relationship between environmental disclosure (ED) and financial performance (FP). This implies that a strong environmental disclosure increases financial performance while a weak one decreases it. Furthermore, the study suggests a moderating effect of social and ethical practices in the link between environmental disclosure and the firm's financial performance. In fact, these findings provide interesting insights for academic practitioners and regulators who are interested in discovering environmental disclosure, firm's performance, and social and ethical practices. These findings also provide insights to stakeholders and regulators on the crucial need to integrate more social and environmental regulations to promote sustainability. Moreover, this paper fills the gaps existing in previous studies that ignore the moderating role of social and ethical practices in the relationship between environmental disclosure and financial performance.

**Keywords:** financial performance; environmental disclosure; environmental, social and governance (ESG); business ethics

## 1. Introduction

Corporate environmental responsibilities have become an international trend. Moreover, environmental strategies have increasingly become a part of business practice and research [1]. The world has experienced extreme environmental degradation, which has profoundly affected people's normal lives. Therefore, the emergence of social responsibility and firm's sustainability puts managers and the old "shareholder–manager–creditor" vision under pressure from the corporate social environment. By showing the existing interconnections between nonfinancial and financial information, environmental disclosure as a strategy seeks long-term improvements in corporate competitiveness [2]. In fact, there is a growing need for companies to provide ED and improve their FP [3]. As such, activity reports must account for, and cite all decisions and consequences of, a company's activity affecting some or all stakeholders.

Although sustainability reports are largely descriptive in nature, they can emerge as an actual complement to exceptional communications. Regarding this motivation, liter-

ature on the determinants of the financial performance (FP) have attempted to present the environmental disclosure as a determinant of this performance [4]. In line with the contributions related to this paper, financial reporting presently suffers from lack of credibility. Therefore, greater credence is typically being given to social and environmental engagement. For this reason, ED can generate a financial performance for shareholders. In fact, an index of ESG firms (Environmental, Social, and Governance), which is used as a proxy of firms' engagement in social and ethical action, has been developed by ASSET4® database of Data-Stream [5].

In this perspective, social and ethical practices have currently attracted growing attention as a worldwide promoter due to its contributions to environmental protection and firm financial performance. Thus, social and moral practices are increasingly introducing adjustments into the marketplace. These practices may impair the firm's financial performance, especially if traders view such engagement as a "cheap talk". Based on this analysis, previous studies have shown that a company's strengths with reference to social and ethical factors have an effect on its FP. This situation opens up opportunities for ESG firms with the presence of environmental problems.

Thus, despite the theoretical support, the moderating effect of social and ethical practices in the link between ED and FP is still much unexplored within the current literature. In fact, most of these studies have also focused on the direct effect in this relationship. However, the contributions concerning the determinants of financial performance are limited. Therefore, the significant gap in literature makes it necessary to analyze the moderating effect of social and ethical practices that can explain the link between ED and FP. Therefore, the purpose of this paper was to bridge this gap by analyzing the moderating effects of social and ethical practices on ED and FP. Thus, we can summarize our problem with the following research questions:

(1) How could environmental disclosure influence the firm's financial performance?
(2) How do social and ethical practices reinforce the relationship between ED and FP?

The purpose of the current paper is to investigate the relationship between social performance and FP, and the moderating effect of social and ethical practices on this relationship.

To test the study's hypotheses, the authors applied linear regressions with a panel data using the Thomson Reuters ASSET4 and Bloomberg database in analyzing data of 523 listed companies selected from the ESG index between 2005 and 2019.

The empirical results indicate a growing interest in corporate social responsibility and information environmental disclosure. Our findings indicate that integrating ED into corporate strategy is an asset that guarantees FP. This disclosure is considered as a means to reduce the information asymmetry through facilitating strategic and financial analysis.

Our research offers the information user a vision to better assess the future growth opportunities in a context where the corporate environmental responsibilities occupies a central position in financial performance. Moreover, this paper fills the gaps existing in previous studies that ignore the moderating role of social and ethical practices in the relationship between environmental disclosure and financial performance [6]. To fill this gap in the literature, we explored the moderating effect of CSR practices and ethical behaviors. Therefore, this paper analyzes whether the interaction between environmental disclosure and social ethical behaviors and practices affects financial performance.

We also contribute to the literature on environmental disclosure, especially in the context of ESG companies, by examining how and to what extent companies use practices related to corporate social responsibility and ethical behavior to improve financial performance.

Furthermore, the importance of this study stems from the fact that it offers important contributions to social and ethical literature and also highlight a growing frame of literature dealing with the effect of sustainability engagement in the firm's financial performance. In addition, our results provide insights to the existing literature on the new creative financial performance strategies, mainly within the context of ESG corporations.

Our analysis is structured as follows: Section 2 presents a literature review and outlines the research hypotheses. Section 3 discusses the sample selection and its characteristics, such as the data sources and the model specification. Then, the main empirical results are presented and discussed in Section 4. Then, the analyses are undertaken in Section 5. Section 6 shows the discussion of findings and our research contributions and implications. Finally, the paper ends with some concluding remarks outlined in Section 7.

## 2. Relevant Prior Literature and Hypotheses Development

This paper examines financial performance with regard to environmental performance disclosure. In this context, the study examines the moderating effect of social and ethical practices taking into account the link between environmental disclosure and the firm's financial performance. In this sphere, sustainability is defined as a concept by which companies integrate social and environmental practices into their business strategy. We look into the ESG firm's ability to provide new insights on the impact of the environmental information disclosure on FP and the impact of social and ethical practices on this relationship.

### 2.1. Hypothesis Development (H1): ED and FP

Company performance has primarily focused on commercial objectives with less attention to social aspects that have not been sufficiently identified as having an exquisite ability. However, the link between ED and FP is controversial [6]. The issue of how environmental disclosure affects a firm's financial performance has been the subject of contentious debate. Research on this topic nonetheless continues to be scarce, which reveals the conduction of latest researches to better apprehend the field and to fill in this gap. In fact, the scope of this research gap consists in regarding the association between ED and FP in ESG companies. In this view, Jabbour et al. [7] found a positive relationship between disclosure of socio-environmental information and market value. However, Matsumura et al. [8] stipulate that data about inefficiency and environmental tasks might also weaken company competition and FP and, hence, can drastically raise the litigation fee.

On the other hand, some studies indicate that sustainability policy can impress FP of firms [9]. In this regard, previous research has stated the importance of CSR practices as an indicator of overall performance and proper management for the various companions of the company [10]. Similarly, existing studies on environmental accounting reveal that information disclosure plays a critical role within the company's performance with regard to pollutant reduction and pollutant emissions [11]. This result suggests that the marketplace is becoming increasingly annoyed with regards to non-financial information, which shows that the disclosure of this fact can bring competitive blessings. In the context of the determinants of performance, Garcia et al. [12] stipulate that ED is prominent for international companies involved in social obligation and sustainability practices. Thus, given the extant findings, we assume an association between the companies' ED and its market value.

Signal and legitimacy theories also have some ideas to offer regarding information disclosure and the effect of environmental engagement on the financial performance. The information disclosure is a requirement of the marketplace as a consequence of the informational asymmetry that takes place between internal and outside users. Therefore, there are wide concerns over the determinants and effects of environmental, social, and governance (ESG) practices in corporate performance [13].

Accordingly, the enterprise engages in win-win good judgment and considers that ED is a sustainable advantage in a turbulent environment. Thus, the first hypothesis is as follows:

**Hypothesis 1 (H1).** *There is a positive association between ED and FP.*

*2.2. Hypotheses Development (H2a and H3a): ED and FP: The Moderating Role of Social and Ethical Practices*

In view of previous literature, social and ethical practices can be a strategic moderator among the relationship between ED and FP. This inevitably leads to an improvement in information disclosure and greater attention to firm financial performance.

**The moderating effect of ethical behavior.** Recently, ethical behavior has increasingly received more attention in terms of its relation to FP and information disclosure. ESG firms values the importance of the ethical engagement to reflect their faithful image towards all the stakeholders and values the importance of transparent and reliable communication [14]. Several authors claim that business ethics rooted within the company way of life has a tremendous impact on the FP and ED [15]. In different phrases, to restrict the problems of agencies, corporate executives are advocated to strengthen sustainability practices based on moral concepts. Thus, the increased attention and increased environmental monitoring lead the company to improve the quality of environmental disclosure. For that reason, ED being consistent with ethical concepts, is considered as a primary solution for corporations wishing to be extra competitive. Those authors show that firms used ethical behavior to fight against corruption and contributing to FP. Thus, acting ethically has many benefits, such as improving firm financial performance [16–18].

Given that the previous debate presents reasons that endorse positive ethical behavior moderating positions, we suggest the following hypotheses:

**Hypothesis 2a (H2a).** *Ethical behavior is positively related to firm's FP.*

**Hypothesis 2b (H2b).** *The link between ED and firm's FP will be positively moderated by ethical behavior.*

**The moderating effect of CSR.** In recent years, CSR practices and its effect on FP are considered as a relatively new area of inquiry. Therefore, the main notion of company social responsibility is that it could be a first-rate tool that aims to reinforce a firm's financial performance. This type of social practices is increasingly important for company reputation. Vourvachis et al. [19] postulate that CSR is considered as a crucial subject matter in many fields. Additionally, according to Garcia et al. [15] and Karim et al. [20], several primarily market-based incentives, were initiated to inspire corporations to conduct ED.

To assist in this final thought, CSR would turn out to be more and more important, which means the creation of financial value for society and its environment. Thus, several theories can explain the relationship between the goals of companies and society. Inside the view of the scope of application of legitimacy theory and signal theory, the company's social responsibility has the capability to boom the overvaluation of the stock market value of the corporation [12].

Literature has proven that company social obligation can undoubtedly impact both businesses' performance. Indeed, ED with CSR practices can lead to a growth in income and facilitate to obtain entry to the credit market with lower interest quotes. Thus, due to the importance of the topic and in response to stock markets on sustainable practices, rating agencies and information providers insist on being given, and value, the environmental and social information [13]. Therefore, while it is expected that firms would choose to increase ED, they may be convinced that the advantages of environmental data disclosure offset or maybe exceed the related costs [14]. Thus, the cost of ED may outweigh the gain. In this perspective, CSR practices have attracted people's attention, which has affected purchaser's behavior. This inevitably leads to an improvement in firm financial performance. Hence, the hypothesis is suggested as follows:

**Hypothesis 3a (H3a).** *Corporate social responsibility is positively related to firm's FP.*

**Hypothesis 3b (H3b).** *The link between ED and firm's FP will be positively moderated by corporate social responsibility.*

## 3. Research Design

### 3.1. Sample and Data Sources

The sample firms are first expected to have social responsibility practices and that are part of the ESG ASSET4 database during the period 2005–2019. This paper uses two kinds of big data sources. The first data are extracted from the "ASSET4 database (Data Stream)". The second source of data is derived from the Bloomberg database. The initial sample of this paper includes 1562 companies. We exclude firms that miss the necessary data for the variables used in our analysis. In a second step, financial institutions and insurance companies were eliminated from the initial sample, considering the very peculiar rules that govern these industries. We obtained a final sample of 523, totaling 7845 firm-year observations. The sample selection is captured in the following Table 1.

**Table 1.** The Procedure for Selecting the Sample.

| Panel A: Sample selection | | |
|---|---|---|
| *Sample* | *Firms* | *Observations* |
| Initial sample | 1562 | 23,430 |
| Less companies with missing data | (939) | (14,085) |
| Less financial services firms | (100) | (1500) |
| Final sample | 523 | 7845 |
| **Panel B: Sample distribution by country** | | |
| *Country* | *Firms* | *%* |
| France | 78 | 14.91 |
| Denmark | 25 | 4.78 |
| Sweden | 61 | 11.67 |
| Spain | 35 | 6.69 |
| Germany | 74 | 14.15 |
| UK | 115 | 21.99 |
| Canada | 135 | 25.81 |
| Total | 523 | 100 |
| **Panel C: Distribution by Industry** | | |
| *SIC Industry Division* | *Firms* | *%* |
| Agriculture, Forestry and Fishing | 29 | 5.54 |
| Mining | 54 | 10.32 |
| Construction | 47 | 8.98 |
| Manufacturing | 42 | 8.05 |
| Transportation, Communications, Electric, Gas, and Sanitary service | 71 | 13.57 |
| Wholesale Trade | 57 | 10.89 |
| Retail Trade | 45 | 8.61 |
| Estate | 38 | 7.26 |
| Services | 54 | 10.32 |
| Non-classifiable | 86 | 16.45 |
| Total | 523 | 100 |

Notes: Observations are total of firm-year observations by country and by industry.

### 3.2. Variables' Measurement

Our methodological approach is realized by a measurement of the variables, which will be followed by a presentation of the model to test the hypotheses of the study.

**Dependent variable:**

***Firm financial performance.*** FP information was acquired from many ratios and databases. In this perspective, many accounting and financial ratios: market-to-book value, ROA, ROE, and Tobin's q (TOBINQ) are typically employed as a proxy of FP in environmental management literature [19,20]. This study selects Tobin's Q to measure firm performance.

**Variables of interest:**

*Environmental disclosure.* Based on previous studies on environmental information disclosure, there are many measures used for this variable in different contexts. Previous studies used binary variables to account for the disclosure [21,22]. Other measures are extracted from the Bloomberg database. In the present study, we are referring to the studies of Yu et al. [23] who used the Bloomberg database to measure the disclosure of environmental information by a score that reflects the disclosure of this information [24]. The ED score provided by Bloomberg is based on the available information in the annual reports, corporate social responsibility reports, and firms' websites.

*Corporate social responsibility.* Much recent research has confirmed that the CSR score determined and calculated by the ASSET4 is a valid measurement of corporate social responsibility practices. The CSR score not only reflects past social performance but also represents the firm's future development expectations. In this paper, corporate social responsibility «CSRSCO» is measured by a score determined by the ASSET4 database to ensure comparability between companies.

*Ethical Behavior.* The measurement of ethics is differently operational in different contexts; this is justified by the fact that this concept is more complicated since it relates to several non-measurable dimensions. In this study, we measured this variable by a score developed by ASSET4. The ethical behavior of the company «ETHSCO» is a score that consists of a series of items objectively quantifying the company's ethical performance and its compliance with professional ethics standards.

*Control variables.* Following previous research and working on the improvement of the goodness of the regression model, we have included the following control variables.

- *Leverage (LEV):* is measured by total debt reported to total assets. Thus, following previous research, companies which have higher debt-to-fairness ratios are not efficient at developing value [25]. Firms with high financial leverage will pressure to take measures which include green innovation to meet the necessities of stakeholders for sustainable improvement.

- *Auditor specialization (AUDSPE):* Based on an extensive literature, we measured auditor specialization by a binary variable that takes the value 1 if the auditor is a specialist in the company same activity sector, and 0 otherwise [26]. For this reason, numerous studies also recommend that auditor specialization may be used to reduce the asymmetry problems of the information [27,28].

- *Sustainable development (SUSDEV):* based on an extensive literature, sustainable development is a binary variable that takes the value 1 if the company has a sustainable development committee, and 0 otherwise.

- *Firm size (SIZE):* Prior research assumed that organizational size is an important determinant of firm-level environmental sustainability conduct [29,30]. This variable is the natural log of total assets. We find that larger firms are likely to have more transparency than smaller firms.

- *Legal system (LEGSYST):* is a binary variable that takes the value of 1 if the company belongs to the Anglo-Saxon legal system, and 0 otherwise [31].

A summary of the respected variables is included in Table 2.

**Table 2.** Variables of the study.

| Symbols | Variables | Measures | Source |
|---|---|---|---|
| TOBINQ | Financial performance | Tobin's q ((Book value of assets − book value of equity − deferred taxes + market value of equity)/Book value of assets). | Thomson Reuters ASSET4 (Datastream) |
| EDISC | Environmental disclosure | It is a score developed by Bloomberg database. | Bloomberg database |
| ETHSCO | Ethical Behavior of company. | It is a score developed by Thomson Reuters ASSET4. | Thomson Reuters ASSET4 (Datastream) |
| CSRSCO | Corporate social responsibility. | It is a score developed by Thomson Reuters ASSET4. | Thomson Reuters ASSET4 (Datastream) |
| LEV | Leverage | Leverage (total debt to market value of equity). | Thomson Reuters ASSET4 (Datastream) |
| AUDSPEC | Auditor specialization | A binary variable that takes the value 1 if the auditor is a specialist in same activity sector, and 0 otherwise. | Thomson Reuters ASSET4 (Datastream) |
| LEGSYST | Legal system. | Is a binary variable that takes the value of 1 if the company belongs to the Anglo-Saxon legal system, and 0 otherwise. | Thomson Reuters ASSET4 (Datastream) |
| SUSDEV | Sustainable development | A binary variable that bears the value 1 if the company has a sustainable development committee, and 0 otherwise. | Thomson Reuters ASSET4 (Datastream) |
| SIZE | Firm size | Is the natural logarithm of total assets. | Thomson Reuters ASSET4 (Datastream) |

Notes: This table reports the definitions of the variables used in our study.

**Regression model specification:**

We opt for implementing the multiple linear regression approach. This paper uses forth regression models to test the hypotheses as follows:

$$TOBINQ_{it} = \beta_0 + \beta_1 EDISC_{it} + \beta_2 LEV_{it} + \beta_3 AUDSPE_{it} + \beta_4 SUSDEV_{it} + \beta_5 SIZE_{it} + \beta_6 LEGSYS_{it} + \sum_{j=7}^{21} \beta_j YEAR_{i,t} + \sum_{k=22}^{28} \beta_K COUNTRY_{i,t} + \varepsilon_{it}$$

(**Model 1**)

$$TOBINQ_{it} = \beta_0 + \beta_1 EDISC_{it} + \beta_2 ETHSCO_{it} + \beta_3 CSRSCO + \beta_4 LEV_{it} + \beta_5 AUDSPE_{it} + \beta_6 SUSDEV_{it} + \beta_7 SIZE_{it} + \beta_8 LEGSYS_{it} + \sum_{j=9}^{23} \beta_j YEAR_{i,t} + \sum_{k=24}^{30} \beta_K COUNTRY_{i,t} + \varepsilon_{it}$$

(**Model 2**)

$$TOBINQ_{it} = \beta_0 + \beta_1 EDISC_{it} + \beta_2 ETHSCO_{it} + \beta_3 EDISC_{it} * ETHSCO_{it} + \beta_4 LEV_{it} + \beta_5 AUDSPE_{it} + \beta_6 SUSDEV_{it} + \beta_7 SIZE_{it} + \beta_8 LEGSYS_{it} + \sum_{j=9}^{23} \beta_j YEAR_{i,t} + \sum_{k=24}^{30} \beta_K COUNTRY_{i,t} + \varepsilon_{it}$$

(**Model 3**)

$$TOBINQ_{it} = \beta_0 + \beta_1 EDISC_{it} + \beta_2 CSRSCO + \beta_3 EDISC_{it} * CSRSCO_{it} + \beta_4 LEV_{it} + \beta_5 AUDSPE_{it} + \beta_6 SUSDEV_{it} + \beta_7 SIZE_{it} + \beta_8 LEG\_SYS_{it} + \sum_{j=9}^{23} \beta_j YEAR_{i,t} + \sum_{k=24}^{30} \beta_K COUNTRY_{i,t} + \varepsilon_{it}$$

(**Model 4**)

where: Year and country stand respectively for year and country fixed effects.

## 4. Research Results

### 4.1. Descriptive Statistics and Correlation Results

Table 3 provides descriptive statistics of variables included in our research models.

**Table 3.** Descriptive statistics.

| Variables | Mean | Q1 | Min | Q2 | Max | SD |
|---|---|---|---|---|---|---|
| *Panel A: Continuous variables* | | | | | | |
| TOBINQ | 1.617 | 1.248 | 1.246 | 2.193 | 2.458 | 1.427 |
| EDISC | 0.625 | 0.3417 | 0.0187 | 0.9101 | 0.9721 | 0.297 |
| CSRSCO | 0.632 | 0.291 | 0.039 | 0.384 | 0.997 | 0.291 |
| ETHSCO | 0.412 | 0.054 | 0.153 | 0.516 | 0.534 | 0.054 |
| LEV | 0.396 | 0.085 | 0.014 | 0.557 | 0.670 | 0.140 |
| SIZE | 6.147 | 4.172 | 3.621 | 7.412 | 8.416 | 1.652 |
| *Panel B: Binary variables* | | | | | | |
| *Variables* | | *Modalities* | | | *%* | |
| SUSDEV | | 0 | | | 49% | |
| | | 1 | | | 51% | |
| AUDSPEC | | 0 | | | 29% | |
| | | 1 | | | 71% | |
| LEGSYS | | 0 | | | 53% | |
| | | 1 | | | 47% | |

Note: This table reports descriptive statistics. Variables definitions are provided in Table 2.

The dependent variable, represented by financial performance (TOBINQ) has an average value of 1.617, with a standard deviation of 1.427 that is very small compared to the average. This variable varies from 1.246 to 2.458, which shows a fairly high ESG firm's financial performance among firms in the world. The firms in the sample have an average environmental disclosure equal to 0.625, which corresponds to an acceptable level of transparency ESG firms. The average of the ethical behavior of the analyzed companies is around (0.412), with a relatively low standard deviation (0.054) compared to the average rate. This fashion is defined by way of the willingness of the ESG businesses to set an approach that promotes moral activities. ESG firms in the sample have an average social responsibility of 0.632, which entails the remarkable dedication of these corporations in socially accountable practices. The mean value of SUSDEV is 51% and 71% for the AUDSPEC.

Clearly, the results show that the number of companies belonging to the Franco-German legal systems is relatively superior to that of those belonging to the Anglo-Saxon legal system (53% of the companies in the sample belong to the legal system of civil law, and 47% are part of the common law legal system).

Table 4 presents Pearson correlation matrix for independent and control variables. This result demonstrates well that no correlation is discovered to be superior to "0.9" [32]. The highest correlation is between the two variables, environmental disclosure and corporate social responsibility (0.8280). This suggests that multicollinearity is not a serious problem.

### 4.2. Panel Unit Root Test

Panel unit root test can be implemented in data analysis to test the stationarity properties of each of the variables and check whether each series is integrated and contains a unit root. We implement two panel unit root tests (Levin-Lin-Chu: LLC and Fisher-types: ADF tests) proposed by Levin et al. [33]. The null hypothesis of the above unit root tests shows that there exists unit root in the series. Rejecting the null hypothesis means the series is stationary. Table 5 shows the results of the panel unit root tests for each variable by applying different unit rout test methods. It can be seen that all variables are stationary at the 1% level (for all tests listed $p < 0.01$), which means no unit root exists in the series.

The results strongly reject the null hypothesis of unit root. Therefore, we can argue that the data are stable, and there is no biased information in the panel.

**Table 4.** Correlation matrix.

| | Variables | 1 | 2 | 3 | 4 | 5 | 6 | 7 | 8 | 9 | 10 |
|---|---|---|---|---|---|---|---|---|---|---|---|
| 1 | EDISC | 1.0000 | | | | | | | | | |
| 2 | ETHSCO | −0.1410 | 1.0000 | | | | | | | | |
| 3 | CSRSCO | 0.8280 | −0.1181 | 1.0000 | | | | | | | |
| 4 | EDISC × ETHSCO | 0.7517 | 0.1656 | 0.7879 | 1.0000 | | | | | | |
| 5 | EDISC × CSRSOC | 0.6403 | −0.1451 | 0.5280 | 0.3901 | 1.0000 | | | | | |
| 6 | LEV | 0.0052 | −0.0323 | 0.0065 | −0.0045 | −0.0007 | 1.0000 | | | | |
| 7 | SIZE | 0.1657 | 0.0015 | 0.2121 | 0.1643 | 0.1802 | 0.1097 | 1.0000 | | | |
| 8 | SUSDEV | 0.3691 | 0.3299 | 0.3914 | 0.3299 | 0.3914 | −0.0442 | 0.0117 | 1.0000 | | |
| 9 | AUDSPEC | 0.1930 | −0.0170 | 0.2260 | 0.1870 | 0.2153 | 0.0079 | 0.1969 | 0.1408 | 1.0000 | |
| 10 | LEGSYS | −0.4351 | 0.1271 | −0.4079 | 0.3937 | −0.4641 | 0.0497 | 0.4400 | −0.2895 | 0.0242 | 1.0000 |

Notes: This table presents the correlation coefficients for variables.

**Table 5.** Panel unit root tests.

| Variable/Method | LLC | | ADF (Fisher Chi-Square) | |
|---|---|---|---|---|
| | No Trend | Trend | No Trend | Trend |
| EDISC | 6.514 *** (0.000) | 3.361 *** (0.000) | 17.178 *** (0.000) | 31.879 *** (0.000) |
| ETHSCO | 12.184 *** (0.000) | 13.710 *** (0.000) | 12.861 *** (0.000) | 11.917 *** (0.000) |
| CSRSCO | 19.383 *** (0.000) | 12.014 *** (0.000) | 23.218 *** (0.000) | 14.878 *** (0.000) |
| EDISC × ETHSCO | 11.297 *** (0.000) | 6.291 *** (0.000) | 15.138 *** (0.000) | 3.752 *** (0.000) |
| EDISC × CSRSOC | 6.871 *** (0.000) | 8.751 *** (0.000) | 17.375 *** (0.000) | 13.395 *** (0.000) |
| LEV | −18.105 *** (0.000) | −13.589 *** (0.000) | 11.271 *** (0.000) | 15.259 *** (0.000) |
| SIZE | 15.678 *** (0.000) | 18.871 *** (0.000) | 18.641 *** (0.000) | 13.972 *** (0.0000) |
| SUSDEV | 7.715 *** (0.000) | 9.271 *** (0.000) | 6.794 *** (0.000) | 7.258 *** (0.000) |
| AUDSPEC | 6.105 *** (0.000) | 10.309 *** (0.000) | 5.917 *** (0.000) | 4.486 *** (0.000) |
| LEGSYS | 2.171 *** (0.001) | 6.124 *** (0.000) | 5.832 *** (0.009) | 6.745 *** (0.015) |

Notes: The statistics in the first row represent the estimated coefficients of the variables while the second row in parentheses represents their respective *p*-values. *** indicate statistical significance at the level of 1%.

### 4.3. Fixed or Random Effect

The decision of fixed and random effect lies on the result of Hausman's test. This test presents a significant result which proves the use of fixed effect regression analysis.

### 4.4. Results of Regression Model

**Step 1:** Check the relationship between ED and FP:

Table 6 shows the regression analysis results associated with Model (1) to test our hypothesis H1 that test the relationship between ED and FP measured by the TOBINQ. The analysis of regression showed that environmental disclosure (EDISC) positively affects the FP of ESG firms. Consequently, the decision to adopt an environmental transparency strategy leads to higher FP. The findings of the study reveal that environmental information disclosure enhances firm performance, which can also be explained by stakeholder theory.

**Table 6.** Regression Results.

| | Model (1) | | Model (2) | | Model (3) | | Model (4) | |
|---|---|---|---|---|---|---|---|---|
| | *Coef* | *t-Statistic* | *Coef* | *t-Statistic* | *Coef* | *t-Statistic* | *Coef* | *t-Statistic* |
| *Intercept* | 3.006 *** | 41.35 | 2.717 *** | 12.86 | 0.224 *** | 5.84 | 2.728 *** | 27.23 |
| EDISC | 0.434 *** | 4.93 | 1.539 *** | 12.27 | 2.593 ** | 1.92 | 1.275 *** | 5.99 |
| ETHSCO | - | - | 0.222 *** | 12.30 | 5.329 ** | 2.33 | - | - |
| CSRSCO | - | - | 1.580 *** | 6.25 | - | - | 1.821 *** | 9.06 |
| EDISC × ETHSCO | - | - | - | - | 5.794 ** | 2.25 | - | - |
| EDISC × CSRSOC | - | - | - | - | - | - | 0.272 *** | 6.26 |
| LEV | −0.292 *** | −6.64 | −0.285 ** | −2.79 | −0.458 ** | 2.41 | −0.031 ** | 1.97 |
| SIZE | 0.411 *** | 5.88 | 0.046 *** | 2.91 | 0.101 *** | 3.00 | 0.733 ** | 2.43 |
| SUSDEV | 1.641 ** | 2.21 | 0.426 *** | −6.15 | 0.030 ** | 2.01 | 0.004 ** | 1.10 |
| AUDSPEC | 0.283 *** | 2.97 | 0.190 ** | 1.98 | 0.055 ** | 2.02 | 0.146 ** | 2.02 |
| LEGSYS | 0.085 ** | 2.04 | 0.043 *** | 3.03 | 0.190 ** | 3.38 | 0.055 * | 1.92 |
| *Firm fixed effects* | Yes | | Yes | | Yes | | Yes | |
| *Year fixed effects* | Yes | | Yes | | Yes | | Yes | |
| *R-squared* | 0.1897 | | 0.2403 | | 0.2973 | | 0.2846 | |
| *F-statistic* | 21.76 *** | | 35.65 *** | | 35.92 *** | | 17.02 *** | |
| *N-obs* | 7845 | | 7845 | | 7845 | | 7845 | |

Notes: Variables definitions are provided in Table 2. The asterisks ***, ** and * indicate significance at the level of 1%, 5%, and 10% respectively.

With respect to the control variables introduced in our regression model, the results show that firm size, sustainable development, and auditor specialization are statistically significant in the explanation of the studied phenomenon. Table 6 presents the explanatory power of the model and a summary of the results of the regression.

**Step 2:** Check the relationship between ED, social, and ethical practices and FP:

Table 6 presents the estimation of model 2 (H2a and H3a), which indicates the direct and indirect role of "social and ethical practices", in FP "TOBINQ". The R-squared is 0.2403. This shows that the regression model can explain 24.03% of the variance in financial performance. The results indicate that our hypotheses are accepted. Aligning with prior research, our findings show a positive and significant link between the social and ethical practices and FP in ESG firms. We find that a superior level of social and ethical practices can provide a better access to FP.

**Step 3:** Check the moderating effect of ethical behavior:

Model 3 was generated and based on Models 1 and 2 by introducing the interactive effect between environmental disclosure (EDISC) and ethical behavior (ETHSOC). Regression results indicated that ethical behavior strengthens the relationship between environmental information disclosure and firm performance.

**Step 4:** Check the moderating effect of corporate social responsibility in the relationship between ED and FP:

In step 4, we need to evaluate the effect of ED on the FP by adding the moderator variable of corporate social responsibility. Model 4 was generated and based on Models 1 and 2 by introducing the interactive variable between information disclosure (EDISC) and social responsibility (CSRSOC). The same Table shows that the F-statistic is 17.02 ($p = 0.000$). This supports that the estimated model is statistically significant.

The results from Table 6 indicate that the coefficient of the interaction variable between social responsibility and environmental information (EDISC × CSRSOC) is significant at 1%, which corroborates our initial expectation. This confirms that a CSR engagement moderates the relationship between information disclosure and FP. Thus, firms are encouraged to use more CSR practices as part of their decision making. Firms with robust CSR practices can fulfill the objective of stakeholders and enhance FP.

## 5. Robustness Checks

To examine the robustness of our results, we conducted additional analyses to verify whether the moderating effect of the social and ethical practices remains intact. Our additional analyses relate to:

(1)　Legal system effect.
(2)　Endogeneity.

### 5.1. Legal System Effect

　　We re-estimate regressions (M1, M2, M3, and M4) using the effect of the legal system in the regression of firm financial performance. We have chosen to test the effect of the legal system and the institutional system as a mechanism that affects corporate governance, which makes it represent a regulatory and legal framework affecting the strategy for creating financial performance. To reach this end, we reviewed our model by subdividing our study sample into two sub-samples. The first sub-sample should involve the companies belonging to Anglo-Saxon system (of a number of 250), while the second one encloses the firms involved in a Franco-German system (of a number of 273). We investigate how the social and ethical practices affect the link between ED and FP by exploring this effect in two different legal systems. As indicated in Table 7, the results with the use of two legal systems is very similar to the original multiple regression results shown in Table 6.

**Table 7.** Robustness test.

| Variable | Panel A (Anglo-Saxon System) | | | | Panel B (Franco-German System) | | | |
|---|---|---|---|---|---|---|---|---|
| | Model 1 | Model 2 | Model 3 | Model 4 | Model 1 | Model 2 | Model 3 | Model 4 |
| Intercept | 1.527 *** (3.08) | 1.761 *** (2.98) | 11.138 ** (1.98) | 1.131 * (1.819) | 1.119 ** (1.98) | 1.121 *** (2.97) | 9.0183 *** (2.66) | 11.076 ** (2.79) |
| EDISC | 0.121 ** (2.11) | 0.131 ** (2.10) | 3.192 *** (3.24) | 3.146 ** (1.986) | 1.102 ** (2.21) | 0.091 *** (2.20) | 1.443 ** (2.08) | 15.08 (4.37) |
| ETHSCO | - | 0.001 ** (2.33) | 8.368 *** (3.01) | | - | 0.352 *** (6.85) | 3.839 *** (7.57) | - |
| CSRSCO | - | 0.117 * (1.81) | - | 4.032 *** (3.13) | - | 0.1987 *** (11.28) | - | 2.32 ** (2.29) |
| EDISC × ETHSCO | - | - | 0.01 * (1.70) | - | - | - | 0.0178 ** (4.58) | - |
| EDISC × CSRSOC | - | - | - | 2.256 *** (4.01) | - | - | - | 3.103 *** (3.15) |
| LEV | −0.192 *** (−4.37) | −0.119 *** (−2.88) | −0.067 *** (−4.08) | −0.043 * (−1.94) | −0.259 *** (−25.45) | −0.154 ** (−2.02) | −1.076 *** (−5.18) | −1.024 ** (−2.25) |
| SIZE | 0.141 * (1.67) | 0.129 *** (3.11) | 4.585 (4.07) | 0.120 ** (1.96) | 0.120 ** (2.06) | 0.163 *** (2.97) | 0.157 ** (2.28) | 0.187 ** (7.24) |
| SUSDEV | 2.568 *** (3.11) | 0.110 * (1.79) | 0.003 *** (5.45) | 0.021 *** (9.36) | 0.091 *** (2.20) | 1.102 ** (2.21) | 0.01 * (1.78) | 0.0457 ** (2.34) |
| AUDSPEC | 1.233 ** (2.07) | 0.122 ** (2.11) | 0.088 *** (6.02) | 0.141 ** (2.21) | 0.111 *** (2.88) | 0.119 *** (2.88) | 0.107 ** (2.27) | 0.026 *** (6.85) |
| *FE Firm* | Yes | Yes | Yes | Yes | Yes | Yes | Yes | Yes |
| *FE Year* | Yes | Yes | Yes | Yes | Yes | Yes | Yes | Yes |
| *R-squared* | 0.3141 | 0.3309 | 0.2742 | 0.3175 | 0.2978 | 0.2426 | 0.2978 | 0.3041 |
| *F-statistic* | 15.67 (p < 0.01) | 14.81 (p < 0.01) | 15.37 (p < 0.01) | 19.71 (p < 0.01) | 14.93 (p < 0.01) | 13.79 (p < 0.01) | 13.47 (p < 0.01) | 13.58 (p < 0.01) |
| *N-obs* | 3750 | 3750 | 3750 | 3750 | 4095 | 4095 | 4095 | 4095 |

Notes: Variables definitions are provided in Table 2. The asterisks ***, ** and * indicate significance at the level of 1%, 5%, and 10% respectively.

　　We have found that the robustness model, as observed at the level of Anglo-Saxon countries, is more relevant than the other model of the Franco-German system (we notice that R-squared of the Anglo-Saxon model is greater than the R-squared of the Franco-German model). Our results would be very interesting for regulators if we conducted a country-by-country analysis. In addition, our results confirm the conclusions of legal and financial theory, according to which countries whose legal traditions are derived from British common law have developed the financial performance creation process better than those that follow French civil law.

### 5.2. Endogeneity Test

The relationship between corporate social responsibility (CSR) and financial performance is one of mitigation. According to Chouaibi et al. [34], CSR engagement and financial performance may be codetermined, with each affecting the other. In this case, estimating either CSR or firm financial performance regressions may result in endogeneity. In order to address the above endogeneity issues, this study applies the Heckman 2-stage specification to correct the selection bias caused by CSR engagement. Further, this study uses a differencing equation to eliminate the simultaneity problem. To control any endogeneity bias stemming from reverse causality, specifically that firms with higher financial performance might be able to afford or support higher CSR levels, we re-estimate our analysis using the instrumental variable approach. Our findings are presented in Model (5) as shown in Table 8. This table shows the results from probit regressions with CSR_DUMMY as the dependent variable. CSR_DUMMY is an indicator variable defining the CSR firm engagement in the top quartile of the distribution.

**Table 8.** Probit model results.

| Variables | (Model 5) CSR_DUMMY |
|---|---|
| Intercept | 3.178 |
| | (1.53) |
| TOBINQ | 3.518 ** |
| | (1.98) |
| EDISC | 3.125 ** |
| | (1.86) |
| ETHSCO | 3.788 ** |
| | (1.98) |
| LEV | −4.178 *** |
| | (6.71) |
| SIZE | 7.261 * |
| | (1.63) |
| SUSDEV | 0.258 ** |
| | (2.71) |
| AUDSPEC | 0.738 *** |
| | (2.17) |
| LEGSYS | 1.432 *** |
| | (3.78) |
| *FE Firm* | YES |
| *FE Year* | YES |
| *Observation* | 7845 |
| *R-squared* | 0.1932 |

Note: The asterisks ***, ** and * indicate significance at the level of 1%, 5%, and 10%.

First, we implement instrumental variable estimation procedures to check whether our results are endogenously affected by the relationship between corporate social responsibility and company financial performance. Consequently, we use the firm-level initial value of the CSR practices as an instrument. This instrumental variable is very likely to be exogenous to the contemporaneous CSR practices score. We employ a two-step regression to estimate model 5. Firstly, we regress the CSR engagement level on the instrumental variable and all the control variables used in the main regression model (see Model 2). Secondly, we preserve the predicted value of CSR score and fit it into our baseline model. The first-stage regression findings show that the instrumental variable is significantly related to the CSR. Then, we save the predicted value of the CSR practices and use it rather than the CSR score in the regression examining the impact of the CSR engagement on financial performance. We show the 2SLS regression results in model 6 (Table 9). These findings are consistent with our main prediction that CSR participation and financial performance are significantly positively correlated, indicating that endogeneity does not drive our main findings.

**Table 9.** Robustness check results on endogeneity.

| Variables | (2) OLS | (6) 2SLS | (7) HECKMAN |
|---|---|---|---|
| Intercept | 2.717 *** | 11.25 ** | 11.356 |
|  | (12.86) | (2.96) | (1.05) |
| EDISC | 1.539 *** | 3.235 ** | 11.47 |
|  | (12.27) | (2.152) | (0.152) |
| ETHSCO | 0.222 *** | 7.498 *** | 6.732 *** |
|  | (12.30) | (6.23) | (11.78) |
| CSRSCO | 1.580 *** | 3.516 ** |  |
|  | (6.25) | (6.978) |  |
| CSR_DUMMY |  |  | 1.853 *** |
|  |  |  | (5.564) |
| LEV | −0.285 ** | −3.248 * | −1.723 ** |
|  | (2.79) | (−1.77) | (−1.89) |
| SIZE | 0.046 *** | 1.113 ** | 0.133 |
|  | (2.91) | (1.95) | (0.11) |
| SUSDEV | 0.426 *** | 1.137 *** | 0.461 |
|  | (6.15) | (3.60) | (2.158) |
| AUDSPEC | 0.190 ** | 3.157 * | 1.756 ** |
|  | (1.98) | (1.85) | (1.97) |
| LEGSYS | 0.043 *** | 4.105 *** | 0.142 ** |
|  | (3.03) | (4.28) | (2.89) |
| LAMBDA |  |  | −3.013 *** |
|  |  |  | (−3.31) |
| *FE Year* | YES | YES | YES |
| *FE Firm* | YES | YES | YES |
| *Homogeneity test* | F = 3.65 | F = 7.59 | F = 7.85 |
|  | (0.000) | (0.000) | (0.000) |
| *White test* | 13.78 | 11.25 | 11.73 |
|  | (0.000) | (0.000) | (0.000) |
| *Hausman test (Khi-deux)* | FE ($p < 0.05$) | FE ($p < 0.05$) | FE ($p < 0.05$) |
| *R-squared* | 0.2403 | 0.2336 | 0.1915 |
| *Observation* | 7845 | 7845 | 7845 |

Notes: Variables definitions are provided in Table 2. (2) OLS: ordinary least squares, (6) 2SLS: the two-stage least squares and (7) Heckman models. The asterisks ***, ** and * indicate significance at the level of 1%, 5%, and 10%.

Second, we use Heckman's [35] two-stage estimation procedure to solve the potential endogeneity of CSR score. First, we redefine the CSR practices into a dummy variable (CSR_DUMMY) coded one for firm-level CSR practices in the top quartile of the distribution. In the first step, a probit model regression using CSR_DUMMY as the dependent variable is carried out. The explanatory variables in the first-stage probit regression include all the independent and control variables from the baseline model (Table 9, Model 6). Using Heckman's two-stage selection model, we correct the specification for endogeneity and test whether CSR practices increase financial performance. The second-stage regression (Table 9, Model 7) results suggest that the positive relationship between CSR practices and financial performance is maintained. The coefficient on LAMBDA is significant in the second-stage regression, implying that firm CSR practices are significantly associated with financial performance. We find the remaining results to be consistent with our hypotheses.

## 6. Discussion of Findings

Ethical and social practices, which are increasingly becoming a strategic issue for ESG companies, depend upon management decision making. For this reason, we have investigated the direct and indirect links between ED and FP using the moderating role of ethical behavior and corporate social responsibility. The obtained results clearly show that companies that exhibit a high level of social and ethical practices are able to enhance their

FP by a high level of ED. We also found that ESG companies are called to increase strategic orientations and reforms leading to all parties involved in value maximization.

Moreover, the evidence presented in our analysis suggests that the link between EDISC and TOBINQ is significant (Model 1; H1), which validates hypothesis H1. In fact, the possible explanation for this result is that the aim behind the presence of ED within companies is to enforce higher FP coverage for the stakeholders. The evidence revealed that ED immediately impacts FP, which implies that our findings are consistent with those in previous studies, such as the result of Fang et al. [36], Lin et al. [37], and Shi et al. [38]. In fact, these authors found that companies need to grow the depth of ED to increase FP. Consequently, for the aforementioned effects to materialize, excellent environmental information should be disclosed to provide a view of the control decision. In this perspective, the presence of an ED and tiers of environmental tracking induces managers to boom the FP. By minimizing agency costs, the environmental disclosure improves firm's performance. On the other hand, as legitimation strategy states, the information disclosure acting as a resource, and enhances performance too. Such information is delivered in the form of legitimacy and consists mainly of the relational capital of the directorates.

Regarding the model 2 (H2a and H3a), the integration of social and ethical practices into the company's strategy is likely to be a determinant of FP in that it represents a mechanism for urging executives with moral conduct and social responsibility practices. Thus, the regression results corroborate the predictions of H2a and H3a and are also in line with the results of previous studies. Empirical evidence is in line with legitimacy theory and signal theory, indicating that the integration of social and ethical practices are associated with environmental disclosure and therefore improve financial performance. Hence, in a risky environment, social and ethical practices could be perceived as a high quality sign to attract responsible investors. This last finding confirms that social and moral practices are considered as an answer that can be aligned with the economic goals of the corporation. In this context, in view of our outcomes, we can say that the purpose behind this is the tendency that the investors need to overvalue the ESG corporations with excessive social and ethical engagement [39,40]. For this reason, social responsibility policy is interpreted by diverse stakeholders through its consequences on the decisions made within the organization. Moreover, the presence of moral practices is a powerful tool for tracking the moves and the control of administrators in order to minimize environmental costs. Therefore, social and ethical practices can make a contribution to the monitoring of the control by favoring an excellent quality disclosure, reducing agency problems, and increasing FP.

Hypothesis H2b (model 3) asserts that the link between ED and FP is positively moderated by ethical behavior. Accordingly, the positive interaction between ethical behavior and information disclosure increases the FP, which confirms our hypothesis (H2b). Therefore, this finding is in agreement with that of Romolini et al. [41] and Cormier et al. [42]. Thus, for several reasons, ESG company values the need and the importance of reliable communication influencing FP. Our results confirm those found in previous works revealing that information asymmetry generates interest conflicts [43,44]. In this regard, given the extant findings, interaction between ethical behavior and information disclosure is a solution to reduce the interest conflicts. Consequently, the obtained results will enable us to recognize the functioning of business ethics in a company that discloses environmental information. Thus, in the light of the legitimacy theory, the disclosure of environmental information in light of ethical behavior has the potential to increase company financial performance. Due to the information disclosure, the agency theory suggests that there is a high relationship between ethical behavior and financial performance.

As can be seen in the Table 5, model 4 (H3b) shows that the interaction between ED and CSR engagement enables companies to benefit from a higher FP. In this regard, our empirical evidence is consistent with legitimacy theory that CSR is an opportunity to reduce conflicts of interest. Thus, CSR is an excellent tool for strengthening the relationship between environmental disclosure and FP of the ESG companies and developing a faithful image.



Therefore, our findings are consistent with those of previous studies [45,46]. Similarly, the evidence provided by our analysis is in conformity with the postulates of the agency theory. Thus, ESG companies that engage in particular practices related to social obligation disclose greater environmental information, which encourages the organization to improve FP in the long run. On the other hand, corporate social obligation is positively interpreted by the diverse stakeholders via its consequences on the choices made by the enterprise. Our findings may be defined with the aid of the attention of the administrators to the CSR as opposed to the satisfactory of ED.

## 7. Conclusions

The purpose of this study was to fill the gap in the literature by theoretically and empirically investigating the logically plausible relationship between environmental information disclosures practiced by firms listed on the ESG index and their FP, using the moderating effect of social and ethical practices. To the best of our knowledge, this work contributes to the growing literature on the environmental information disclosure and social and ethical practices, providing greater insight into their role as determinants of FP. This study also presents the benefits and possible limitations of the ED. Moreover, several previous studies have found that corporate engagement in socially responsible practices and in non-financial disclosure can contribute to the improvement of the effectiveness and FP. This paper is therefore conceived as an extension of this research subject matter by formulating a conclusion about the moderating effect of the social and ethical engagements on the relationship between ED and FP.

Our results confirm our expectations regarding the impact of some variable of our models on financial performance. We also highlighted the importance of the interaction between the social and ethical policy and non-financial disclosure in the firm performance. Empirical results reveal that social strategies adoption is positively associated with the extent of financial performance. This indicates that ESG companies with environmental information disclosure are more likely to create financial performance. This finding implies that information disclosure can play an important role in legitimizing the company's activities by influencing management behavior to make more CSR and, consequently, improve the corporate image.

The paper contributes to the existing literature in several ways. First of all, this research provides a good reference for deriving the basic role of social and ethical engagements in establishing a good reputation. Second, in view of the benefits related to corporate social responsibility and ethics, ESG companies are called on to pay special attention to the environmental and climate change strategies. Our research offers the information user a better vision about future opportunity growth in a context where the environmental information disclosure occupies a central position in business valuation and in its financial performance. Moreover, this work is conceived as an extension of the research topic, trying to document and present conclusions about ESG firms.

Finally, although the entire research process was as rigorous as possible, there are still some limitations. The primary limitation is linked to the nature of our sample, which only includes companies operating in developed countries. This shortcoming highlights the need for future research. Although we concentrate on the social and ethical practices, it is desirable to include other variables of governance mechanisms that may affect a company's financial performance.

**Author Contributions:** Conceptualization, S.C and J.C.; methodology, D.S.; software, S.C; validation, M.R., D.S. and J.C.; formal analysis, D.S.; investigation, J.C.; resources, M.R.; data curation, S.C.; writing—original draft preparation, D.S.; writing—review and editing, M.R.; visualization, J.C.; supervision, M.R.; project administration, S.C.; funding acquisition, D.S. All authors have read and agreed to the published version of the manuscript.

**Funding:** This research received no external funding.

**Institutional Review Board Statement:** Not applicable.

**Informed Consent Statement:** Not applicable.

**Data Availability Statement:** Not applicable.

**Conflicts of Interest:** The authors declare no conflict of interest.

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
