# Peer review of "Exploring the Moderating Role of Social and Ethical Practices in the Relationship between Environmental Disclosure and Financial Performance: Evidence from ESG Companies"

_sustainability, doi:10.3390/su14010209_

Round 1

Reviewer 1 Report

The article has improved significantly, but the two observations made in the previous review are still not fully resolved:

  • If there are multicollinearity problems, I recommend indicating that all the variables could not be analyzed simultaneously due to this problem as a limitation.
  • The format of the citations and references is not yet in the one used by Sustainability.

Author Response

Dear Sir,

It’s a great pleasure to thank you for your comments regarding my manuscript titled " Exploring the moderating role of social and ethical practices in the relationship between environmental disclosure and financial performance: evidence from ESG companies".

Based on your recommendations, we are conscious of the necessity to re-check through the paper to improve its scope. Hence, we have introduced some modifications to its content in the attempt to make it concise and conform to reviewers’ expectations. 

Below we detail the major rectifications that have been done;

  1. The introduction is reconstructed to be specific and concise, yet informative and indicative of the ambition of the paper.
  2. We have also restructured abstract section and empirical results section. Thus, these sections has been clearly presented and appropriately analyzed.
  3. We have checked all the bibliographical references in our article, and we have reinforced them with articles published in 2020 and 2021.
  4. We also formatted the references according to the MDPI (sustainability) style, and we paid great attention to the punctuation and the order of the references.
  5. Errors, typos and grammar errors have been corrected. We hired an English language specialist to correct the incorrect sentences.
  6. We considered the previous conceptual framework from the literature as a basic background to conduct our theoretical analysis and achieve our objective and we have strengthened our review of the literature with recently published articles.

Reviewer:1
Comments and Suggestions for Authors

The article has improved significantly, but the two observations made in the previous review are still not fully resolved:

Question

  • If there are multicollinearity problems, I recommend indicating that all the variables could not be analyzed simultaneously due to this problem as a limitation.

Response

Based on your recommendations, we have examined if there are multicollinearity problems in the 4 models. The results found show the absence of this problem. Robust models are used the methodology arguments are well developed. We have described more accurately the origin of the variables included in the models, making it easier to understand them. Furthermore, the choices of our variables are supported by the literature. We specified and tested stationarity, multicollinearity and endogeneity of our data.

Question

  • The format of the citations and references is not yet in the one used by Sustainability.

Response

Based on your recommendations great attention has been paid to the format of the document (citations, references, tables, etc.).

Errors, typos and grammar errors have been corrected. We hired an English language specialist to correct the incorrect sentences. We also checked for typos and grammatical errors and corrected them.

Reviewer 2 Report

The topic presented in this work is really interesting. However, several challenges are required:

I analyze the single sections:

Abstract has inappropriate structure. I suggest to answer the following aspects: - general context - novelty of the work - methodology used (describe briefly the main methods or treatments applied) - main results and related interpretations.

Introduction: This section should briefly place the study in a wide context and emphasize why it is relevant carrying out the analysis. It should define the purpose of the work and its significance. In this perspective, this section is too succinct and fails to effectively point out the relevance of your contribution towards the existing literature.

An important point is to introduce the relationship of bank-firm and green investments’ strategies as well as the role of green finance and regulation.

Some reference to start with are: 

https://www.sciencedirect.com/science/article/pii/S0959652620318461

https://www.sciencedirect.com/science/article/pii/S1876610216316113

https://www.mdpi.com/2071-1050/11/2/517

Materials and methods: I found this section very important for the readability of the paper. The research methodology seems underdeveloped. Methods should be described in detail. I think the research procedure could be much more clearly described by means of a diagram also highlighting its potential and limit.

Discussions: The discussion of the results is merely descriptive and the obtained evidence is flimsy due to the fact the outcomes are not supported by an adequate discussion in light of scientific literature. Authors should discuss the results and how they can be interpreted in perspective of previous studies and their implications should be discussed in the broadest context possible.

Conclusions: Conclusions must also be revised according to the previous comments. In particular, they should discuss practical and policy implications as well as future lines of research.

I hope these comments might help in improving the paper and encourage the authors to move forward.

Author Response

Dear Sir,

It’s a great pleasure to thank you for your comments regarding my manuscript titled " Exploring the moderating role of social and ethical practices in the relationship between environmental disclosure and financial performance: evidence from ESG companies".

Based on your recommendations, we are conscious of the necessity to re-check through the paper to improve its scope. Hence, we have introduced some modifications to its content in the attempt to make it concise and conform to reviewers’ expectations. 

Below we detail the major rectifications that have been done;

  1. The introduction is reconstructed to be specific and concise, yet informative and indicative of the ambition of the paper.
  2. We have also restructured abstract section and empirical results section. Thus, these sections has been clearly presented and appropriately analyzed.
  3. We have checked all the bibliographical references in our article, and we have reinforced them with articles published in 2020 and 2021.
  4. We also formatted the references according to the MDPI (sustainability) style, and we paid great attention to the punctuation and the order of the references.
  5. Errors, typos and grammar errors have been corrected. We hired an English language specialist to correct the incorrect sentences.
  6. We considered the previous conceptual framework from the literature as a basic background to conduct our theoretical analysis and achieve our objective and we have strengthened our review of the literature with recently published articles.

Reviewer:2
Comments and Suggestions for Authors

The topic presented in this work is really interesting. However, several challenges are required:

I analyze the single sections:

Question

Abstract has inappropriate structure. I suggest to answer the following aspects: - general context - novelty of the work - methodology used (describe briefly the main methods or treatments applied) - main results and related interpretations.

Response

Abstract is reconstructed to be specific, concise, informative and indicative of the ambition of the paper. Thus, this section has been clearly presented and appropriately analyzed and minimized.

Question

Introduction: This section should briefly place the study in a wide context and emphasize why it is relevant carrying out the analysis. It should define the purpose of the work and its significance. In this perspective, this section is too succinct and fails to effectively point out the relevance of your contribution towards the existing literature.

An important point is to introduce the relationship of bank-firm and green investments’ strategies as well as the role of green finance and regulation.

Response

Based on your recommendations, introduction is reconstructed to be specific, concise, informative and indicative of the ambition of the paper. Thus, this section has been clearly presented and appropriately analyzed and minimized.

We have detailed the main contributions of our article as well as the main results found (Please see page 3). Our research offers the information user a vision to better assess the future growth opportunities in a context where the corporate environmental responsibilities occupies a central position in financial performance.

Question

Some reference to start with are:

https://www.sciencedirect.com/science/article/pii/S0959652620318461

https://www.sciencedirect.com/science/article/pii/S1876610216316113

https://www.mdpi.com/2071-1050/11/2/517

Response

Based on your recommendations, we have strengthened our review of the literature with recently published articles. We considered the previous conceptual framework from the literature as a basic background to conduct our theoretical analysis and achieve our objective and we have strengthened our review of the literature with recently published articles.

Question

Materials and methods: I found this section very important for the readability of the paper. The research methodology seems underdeveloped. Methods should be described in detail. I think the research procedure could be much more clearly described by means of a diagram also highlighting its potential and limit.

 Response

In order to ensure the quality of the results found we have chosen the Thomson Reuters and Bloomberg database. We considered the previous conceptual framework from the literature as a basic background to conduct our methodology analysis and achieve our objective and we have strengthened our review of the literature with recently published articles.

Question

Discussions: The discussion of the results is merely descriptive and the obtained evidence is flimsy due to the fact the outcomes are not supported by an adequate discussion in light of scientific literature. Authors should discuss the results and how they can be interpreted in perspective of previous studies and their implications should be discussed in the broadest context possible.

 Response

Based on your relevant advice regarding the results and to ensure the comparability of our study with previous studies, we compared the results found with those of other studies and we have detailed the main contributions of our article as well as the main results found.

 Question

Conclusions: Conclusions must also be revised according to the previous comments. In particular, they should discuss practical and policy implications as well as future lines of research.

 Response

Conclusion is reconstructed and revised according to the previous comments. In particular, we are discussing our practical implications as well as future lines of research.

I hope these comments might help in improving the paper and encourage the authors to move forward.

Reviewer 3 Report

The paper investigated the relationship betwenn environmental disclosure (ED) practiced by firms listed on the ESG index affects their financial performance (FP). 

It is a paper with a intensely approached topic, a lot has been written on this topic, so I think that more research could be cited.

Please verify the citations and references, it is necessary to be modified according with journal requirements.

https://www.mdpi.com/journal/sustainability/instructions

It must be specified whether the data have been tested, for stationarity, multicollinearity or another applicable test. How did you solve the problem of endogeneity?

Please clarify the following sentences, because in the first is mentioned that results are very similar, and in the second is 'there are majors differences". What are the majors differences?

Lines 357-358

"As indicated in Table 6, the results with the using of two legal system is very similar to the original multiple regression results shown in Table 5." 

Lines 359-350

This result shows that there are major differences in the institutional framework between the two legal systems discussed. 

In the discussion section, a table in which to make a synthesis of the hypotheses and how each was validated might be useful.

Author Response

Reviewer:3

Comments and Suggestions for Authors

The paper investigated the relationship between environmental disclosure (ED) practiced by firms listed on the ESG index affects their financial performance (FP). 

It is a paper with a intensely approached topic, a lot has been written on this topic, so I think that more research could be cited.

Question

Please verify the citations and references, it is necessary to be modified according with journal requirements.

https://www.mdpi.com/journal/sustainability/instructions

Response

Based on your recommendations great attention has been paid to the format of the document (citations, references, tables, etc.). Errors, typos and grammar errors have been corrected. We hired an English language specialist to correct the incorrect sentences. We also checked for typos and grammatical errors and corrected them. We also formatted the references according to the MDPI (sustainability) style, and we paid great attention to the punctuation and the order of the references.

Question

It must be specified whether the data have been tested, for stationarity, multicollinearity or another applicable test. How did you solve the problem of endogeneity?

Response

Based on your recommendations we specified and tested stationarity, multicollinearity and endogeneity of our data (Please see page 15 and 21-22).

Question

Please clarify the following sentences, because in the first is mentioned that results are very similar, and in the second is 'there are majors differences". What are the majors differences?

Lines 357-358

"As indicated in Table 6, the results with the using of two legal system is very similar to the original multiple regression results shown in Table 5." 

Lines 359-350

This result shows that there are major differences in the institutional framework between the two legal systems discussed. 

Response

On the basis of your recommendations, we have clearly specified our robustness tests and we have reinforced our choices and our interpretations.

Question

In the discussion section, a table in which to make a synthesis of the hypotheses and how each was validated might be useful.

Response

Based on your relevant advice regarding the results and to ensure the comparability of our study with previous studies, we compared the results found with those of other studies and we have detailed the main contributions of our article as well as the main results found.

Round 2

Reviewer 2 Report

Thank you for the revised version. Altough the authors struggled to improved the manuscript they did not properly succeed. This is particularly true for the  introduction and method sections. I still believe that literature suggested might help in improving the manuscript. I report it here for your convinience:

An important point is to introduce the relationship of bank-firm and green investments’ strategies as well as the role of green finance and regulation.

Some reference to start with are:

https://www.sciencedirect.com/science/article/pii/S0959652620318461

https://www.sciencedirect.com/science/article/pii/S1876610216316113

https://www.mdpi.com/2071-1050/11/2/517

Author Response

Reviewer:2
Comments and Suggestions for Authors

The topic presented in this work is really interesting. However, several challenges are required:

I analyze the single sections:

Question

Abstract has inappropriate structure. I suggest to answer the following aspects: - general context - novelty of the work - methodology used (describe briefly the main methods or treatments applied) - main results and related interpretations.

Response

Abstract is reconstructed to be specific, concise, informative and indicative of the ambition of the paper. Thus, this section has been clearly presented and appropriately analyzed and minimized.

Question

Introduction: This section should briefly place the study in a wide context and emphasize why it is relevant carrying out the analysis. It should define the purpose of the work and its significance. In this perspective, this section is too succinct and fails to effectively point out the relevance of your contribution towards the existing literature.

An important point is to introduce the relationship of bank-firm and green investments’ strategies as well as the role of green finance and regulation.

Response

Based on your recommendations, introduction is reconstructed to be specific, concise, informative and indicative of the ambition of the paper. Thus, this section has been clearly presented and appropriately analyzed and minimized.

We have detailed the main contributions of our article as well as the main results found (Please see page 3). Our research offers the information user a vision to better assess the future growth opportunities in a context where the corporate environmental responsibilities occupies a central position in financial performance.

Question

Some reference to start with are:

https://www.sciencedirect.com/science/article/pii/S0959652620318461

https://www.sciencedirect.com/science/article/pii/S1876610216316113

https://www.mdpi.com/2071-1050/11/2/517

Response

Based on your recommendations, we have strengthened our review of the literature with recently published articles. We considered the previous conceptual framework from the literature as a basic background to conduct our theoretical analysis and achieve our objective and we have strengthened our review of the literature with recently published articles.

Question

Materials and methods: I found this section very important for the readability of the paper. The research methodology seems underdeveloped. Methods should be described in detail. I think the research procedure could be much more clearly described by means of a diagram also highlighting its potential and limit.

 Response

In order to ensure the quality of the results found we have chosen the Thomson Reuters and Bloomberg database. We considered the previous conceptual framework from the literature as a basic background to conduct our methodology analysis and achieve our objective and we have strengthened our review of the literature with recently published articles.

Question

Discussions: The discussion of the results is merely descriptive and the obtained evidence is flimsy due to the fact the outcomes are not supported by an adequate discussion in light of scientific literature. Authors should discuss the results and how they can be interpreted in perspective of previous studies and their implications should be discussed in the broadest context possible.

 Response

Based on your relevant advice regarding the results and to ensure the comparability of our study with previous studies, we compared the results found with those of other studies and we have detailed the main contributions of our article as well as the main results found.

 Question

Conclusions: Conclusions must also be revised according to the previous comments. In particular, they should discuss practical and policy implications as well as future lines of research.

 Response

Conclusion is reconstructed and revised according to the previous comments. In particular, we are discussing our practical implications as well as future lines of research.

I hope these comments might help in improving the paper and encourage the authors to move forward.

Thank you

Reviewer 3 Report

The paper is improved, but I think there are other issues that need to be clarified.

I saw that you created model 5, does this refer to any hypothesis? 

For clarification, I still consider that a table with 3 columns would be necessary: the code of the hypotheses, the model of each one and their validation.

Author Response

Reviewer:3
I saw that you created model 5, does this refer to any hypothesis? 

Response
It is not a question of a hypothesis but it is a test of robustness

Comments and Suggestions for Authors

The paper investigated the relationship betwenn environmental disclosure (ED) practiced by firms listed on the ESG index affects their financial performance (FP). 

It is a paper with a intensely approached topic, a lot has been written on this topic, so I think that more research could be cited.

Question

Please verify the citations and references, it is necessary to be modified according with journal requirements.

https://www.mdpi.com/journal/sustainability/instructions

Response

Based on your recommendations great attention has been paid to the format of the document (citations, references, tables, etc.). Errors, typos and grammar errors have been corrected. We hired an English language specialist to correct the incorrect sentences. We also checked for typos and grammatical errors and corrected them. We also formatted the references according to the MDPI (sustainability) style, and we paid great attention to the punctuation and the order of the references.

Question

It must be specified whether the data have been tested, for stationarity, multicollinearity or another applicable test. How did you solve the problem of endogeneity?

Response

Based on your recommendations we specified and tested stationarity, multicollinearity and endogeneity of our data (Please see page 15 and 21-22).

Question

Please clarify the following sentences, because in the first is mentioned that results are very similar, and in the second is 'there are majors differences". What are the majors differences?

Lines 357-358

"As indicated in Table 6, the results with the using of two legal system is very similar to the original multiple regression results shown in Table 5." 

Lines 359-350

This result shows that there are major differences in the institutional framework between the two legal systems discussed. 

Response

On the basis of your recommendations, we have clearly specified our robustness tests and we have reinforced our choices and our interpretations.

Question

In the discussion section, a table in which to make a synthesis of the hypotheses and how each was validated might be useful.

Response

Based on your relevant advice regarding the results and to ensure the comparability of our study with previous studies, we compared the results found with those of other studies and we have detailed the main contributions of our article as well as the main results found.

Reviewer:4

Question

  1. Our journal does not force authors to cite all the references that
    reviewers/editors recommended in the review reports. If you think some
    references are unnecessary, you are suggested to provide explanations in the
    author reply to reviewer's comments when you submit the revised manuscript.
    Response

Thank you for your recommendation, in fact in order to strengthen the quality of our article we have taken into consideration all the recommendations given by the rapporteurs.

Question

  1. Please reduce the repetition rate of your paper.
    Response

Thank you for your recommendation; we hired an English language specialist to correct the some sentences and constructions. Also, we have attempted to solve the problem of redundancy in the present paper.

Round 3

Reviewer 2 Report

Thanks for this revised version. After two rounds revision I do not believe the paper has achieved a publishable standard. My comments during these two stage of revision have not been properly considered.

The introduction fails to set out any specific interest to a broader audience. There is nothing more than a sort of putting forward the topic, the claim of a research question and the description of the paper structure. However, what about contribution to relevant literature? The author stated “ this paper offers incentives for managers to invest in ED and in social and ethical practices in their quest for sustainable FP”. Which literature are you looking at, where is the research gap? In this case, it is not sufficiently supported.

The theoretical background for hypothesis definition, is shallow and devoid of insight. Therefore, there is not any theoretical proposition; neither are the authors attempting to support any hypothesis. The cited literature is outdated and scarcely interesting for the journal audience. What about recent research? 

It is undeniably true that the authors are clear in providing their methodological approach. Nevertheless, this research technique seems to have been employed without any rigour. We cannot know how representative the extracted sample is. Similarly, the authors state that they performed a linear regression but we know nothing about how they put into motion this research technique.

It is not assuming a theoretical contribution that matters most but rather providing new insightful doctrines and theories derived from the obtained results. To put it briefly, there are no practical implications for policy. 

I hope these comments might help in improving the paper and encourage the authors to move forward. 

Author Response

Response to Reviewer 2

Point 1: The introduction fails to set out any specific interest to a broader audience. There is nothing more than a sort of putting forward the topic, the claim of a research question and the description of the paper structure. However, what about contribution to relevant literature? The author stated “ this paper offers incentives for managers to invest in ED and in social and ethical practices in their quest for sustainable FP”. Which literature are you looking at, where is the research gap? In this case, it is not sufficiently supported.

 Response 1:

 We thank the reviewer for this comment. Based on your recommendations, we have also restructured introduction section. We have detailed the main contributions of our article as well as the main results found (Please see page 2).

To test the study’s hypotheses, the authors applied linear regressions with a data panel using the Thomson Reuters ASSET4 and Bloomberg database in analyzing data of 523 listed companies selected from the ESG index between 2005 and 2019.

The empirical results indicate a growing interest in corporate social responsibility and information environmental disclosure. Our findings indicate that integrating ED into corporate strategy is an asset that guarantees FP. This disclosure is considered as a means to reduce the information asymmetry through facilitating strategic and financial analysis.

Our research offers the information user a vision to better assess the future growth opportunities in a context where the corporate environmental responsibilities occupies a central position in financial performance. Moreover, this paper fills the gaps existing in previous studies that ignore the moderating role of social and ethical practices in the relationship between environmental disclosure and financial performance [6]. To fill this gap in the literature, we explored the moderating effect of CSR practices and ethical behaviors. Therefore, this paper analyzes whether the interaction between environmental disclosure and social ethical behaviors and practices affects financial performance.

We also contribute to the literature on environmental disclosure, especially in the context of ESG companies, by examining how and to what extent companies use practices related to corporate social responsibility and ethical behavior to improve financial performance.

Furthermore, the importance of this study stems from the fact that it offers important contributions to social and ethical literature and also highlight a growing frame of literature dealing with the effect of sustainability engagement in the firm’s financial performance. Besides, our results provide insights to the existing literature on the new creative financial performance strategies, mainly within the context of ESG corporations.

Point 2: The theoretical background for hypothesis definition, is shallow and devoid of insight. Therefore, there is not any theoretical proposition; neither are the authors attempting to support any hypothesis. The cited literature is outdated and scarcely interesting for the journal audience. What about recent research?

Response 2:

We thank the reviewer for this comment; we have strengthened our review of the literature with recently published articles. We considered the previous conceptual framework from the literature as a basic background to conduct our theoretical analysis and achieve our objective and we have strengthened our review of the literature with recently published articles.

Point 3: It is undeniably true that the authors are clear in providing their methodological approach. Nevertheless, this research technique seems to have been employed without any rigour. We cannot know how representative the extracted sample is. Similarly, the authors state that they performed a linear regression but we know nothing about how they put into motion this research technique.

Response 3:

We thank the reviewer for this comment; we specified and tested stationarity, multicollinearity and endogeneity of our data (Please see page 15 and 21-22), and we have clearly specified our robustness tests and we have reinforced our choices and our interpretations. In the other hand, regarding the results and to ensure the comparability of our study with previous studies, we compared the results found with those of other studies and we have detailed the main contributions of our article as well as the main results found.

We presented the choice of our sample as well as the process of their selection (Please see page 5).

Point 4: It is not assuming a theoretical contribution that matters most but rather providing new insightful doctrines and theories derived from the obtained results. To put it briefly, there are no practical implications for policy.

Response 4:

Based on your recommendations, we have detailed the main contributions of our article as well as the main results found (Please see page 2).

To test the study’s hypotheses, the authors applied linear regressions with a data panel using the Thomson Reuters ASSET4 and Bloomberg database in analyzing data of 523 listed companies selected from the ESG index between 2005 and 2019.

The empirical results indicate a growing interest in corporate social responsibility and information environmental disclosure. Our findings indicate that integrating ED into corporate strategy is an asset that guarantees FP. This disclosure is considered as a means to reduce the information asymmetry through facilitating strategic and financial analysis.

Our research offers the information user a vision to better assess the future growth opportunities in a context where the corporate environmental responsibilities occupies a central position in financial performance. Moreover, this paper fills the gaps existing in previous studies that ignore the moderating role of social and ethical practices in the relationship between environmental disclosure and financial performance [6]. To fill this gap in the literature, we explored the moderating effect of CSR practices and ethical behaviors. Therefore, this paper analyzes whether the interaction between environmental disclosure and social ethical behaviors and practices affects financial performance.

We also contribute to the literature on environmental disclosure, especially in the context of ESG companies, by examining how and to what extent companies use practices related to corporate social responsibility and ethical behavior to improve financial performance.

Furthermore, the importance of this study stems from the fact that it offers important contributions to social and ethical literature and also highlight a growing frame of literature dealing with the effect of sustainability engagement in the firm’s financial performance. Besides, our results provide insights to the existing literature on the new creative financial performance strategies, mainly within the context of ESG corporations.

Point 5: I hope these comments might help in improving the paper and encourage the authors to move forward.

Response 5:

We thank the reviewer for this comment.

This manuscript is a resubmission of an earlier submission. The following is a list of the peer review reports and author responses from that submission.

Round 1

Reviewer 1 Report

Referee report on manuscript sustainability-1434012:

" Exploring the moderating role of social and ethical practices in the relationship between environmental disclosure and financial performance: evidence from ESG companies"

This manuscript explores the moderating role of social and ethical practices in the relationship between environmental disclosure and financial performance. After reviewing this article, we proposed several concerns for this paper shown below:

Comments

  1. The authors should mention more why the moderating role of social and ethical practices in the relationship between environmental disclosure and financial performance would indeed matter for this study since such an issue would be mainly concerned in this study.

  1. The issue that has not been done before might not indicate that such an issue is worthy of investigation. Authors should provide more supporting arguments to persuade readers such an issue is worthy of further study.

  1. After incorporating interaction terms (EDISC*CSRSOC) or (EDISC*ETHSOC) in your models, do you examine whether multicollinearity issues would exist in your models

  1. According to examining the moderating effect in your regression models, we might have to incorporate EDISC, CSRSOC, (EDISC*CSRSOC) or (EDISC*ETHSOC) together instead of separately in your panel data models as shown in Tables 5-6. Besides, authors have to examine whether multicollinearity issues exist in the models after incorporating interaction terms together instead of separately.

  1. The robustness of your models should be concerned since authors employ the traditional panel data (i.e., fixed effect or random effect according to Hausman’s test). Traditional panel data models might be regarded as one of the out-of-date models that have several shortcomings proposed by Petersen (2009)

Reference:

Petersen, M. A. (2009). Estimating standard errors in finance panel data sets: Comparing approaches. The Review of financial studies, 22(1), 435-480.

  1. Why the legal system could be employed as a proxy for the FP should be explained in detail before presenting Table 6. In addition, since the legal system is a binary variable and the proxy of FP is Tobin’q, authors should provide supporting augments about why the legal system could be employed as the proxy of the FP.

  1. Furthermore, the robustness of empirical results mainly refers to that the results would be almost the same after adopting different appropriate models. Authors should make efforts on the robustness of your empirical results by employing different appropriate models since employing different proxy for FP without supporting arguments might persuade readers the robustness of your results.  

  1. Regarding the conclusion section, this article should present your main conclusions engaging in the dialogue with the relevant literature, research implications (theoretical, managerial, or practical implications), and future studies and limitations. The conclusion section indeed should be enhanced.

Reviewer 2 Report

The work done is interesting, but there are some things to comment on:

  • The authors propose four models, but in no case do they simultaneously analyze the direct and moderating effects of the variables. By only incorporating a part in each model, it could be "forcing" the determination of significant effects when it does not really have to be the case. If all the variables are incorporated at the same time, it could be determined whether there is a moderating effect, a direct effect or both exist. Therefore, a Model 5 with all the variables simultaneously would be missing (it could actually be the only model analyzed).
  • The document format is sloppy (citations, references, tables, etc.), some expressions can be improved and the use of terms and acronyms is chaotic, as the first page exemplifies.